# Hierarchical Epidemic Model on Structured Population: Diffusion Patterns and Control Policies

**Elena Gubar** [1,*,†] , **Vladislav Taynitskiy** [1,†] , **Denis Fedyanin** [2,†] and **Ilya Petrov** [3,†]

1   Faculty of Applied Mathematics and Control Processes, Saint Petersburg State University,
    198504 Saint Petersburg, Russia; tainitsky@gmail.com
2   International Laboratory for Logic, Linguistics and Formal Philosophy, National Research University Higher
    School of Economics, 101000 Moscow, Russia; dfedyanin@inbox.ru
3   V. A. Trapeznikov Institute of Control Sciences of RAS, 117997 Moscow, Russia; ilyapetrov22@mail.ru
*   Correspondence: e.gubar@spbu.ru
†   These authors contributed equally to this work.

**Abstract:** In the current study, we define a hierarchical epidemic model that helps to describe the propagation of a pathogen in a clustered human population. The estimation of a novel coronavirus spreading worldwide leads to the idea of the hierarchical structure of the epidemic process. Thus, the propagation process is divided into three possible levels: a city, a country, and a worldwide. On each level, the pathogen propagation process is based on the susceptible-exposed-infected-recovered (SEIR) model. We thus formulate a modified transmission model of infected individuals between levels. The control of the pathogen's spread can be seen as an optimal control problem. A trade-off exists between the cost of active virus propagation and the design of appropriate quarantine measures. Each level of the hierarchy is defined by its network. A series of numerical experiments was conducted to corroborate the obtained results.

**Keywords:** epidemic process; compartment epidemic models; SIR model; optimal control

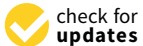



## 1. Introduction

Infectious diseases still pose a serious medical challenge worldwide, with millions of deaths per year estimated to be directly related to infectious diseases. The outbreak of new diseases such as severe acute respiratory syndrome (SARS), Middle East respiratory syndrome (MERS), the rise of Ebola, and, most recently, the COVID-19 pandemic represent a few examples of significant problems that the public healthcare system and medical science research need to address.

Although these sudden epidemics were considered a severe threat for centuries, now there is an ability to control further epidemic outbreaks, which advances in modern science have facilitated. Cures and vaccines for several dangerous pathogens are being developed and manufactured faster than ever before. The vaccines against new influenza A (H1N1) and the novel coronavirus were developed rapidly and made available only a few months after the beginning of the epidemics.

However, one challenge in disease control is that one pathogen can generate many strains with different spreading features. Hence, a detailed investigation of multi-strain epidemic dynamics is of great relevance [1–3]. For instance, the influenza A (H1N1) virus has the potential to develop into the first influenza pandemic of the twenty-first century [4], accompanied by seasonal influenza accompanies [5]. Another important issue is the rate of the pathogen's spread and the network of contacts for each resident in the population. These two facts have been investigated in many of the recent scientific projects [6–8].

Recent studies have seen a surge of interest in using optimal control and stability equilibrium analysis to study epidemic protection in computer networks, social networks, and human populations (see [9–16]). Moreover, dividing the entire population into clusters

plays an important role. Researchers have considered the impact of epidemic outbreaks in different areas [17–20]. One study has presented the optimal countermeasures to resist virus attacks [21]. Previous studies have also shown the application of epidemic frameworks to the models of network protection, as in [7,22–24].

Similarly to our study, researchers have also started incorporating economic trade-offs and conducting optimal control policy analysis within the compartment epidemic model framework, undertaking optimal control analysis in single- and multiple-group models (starting with the early related contributions by [25,26] and recently [1,27–33]).

Although homogeneous population models are classic and have made a significant contribution to understanding the spread of infection and counteracting it, researchers are increasingly paying attention to the heterogeneous nature of the population undergoing the emergence of a new infection. In real life, the interaction between people is of a network nature, and groups of people can differ in terms of various demographic, social, economic, and other factors. In our model, subpopulations are determined based on geographic factors. Disease-related factors such as the average number of contacts, latent period, and probability of an infectious individual recovering are independent and equal for all layers. Such heterogeneity for populations has long been the subject of research; ref. [34] can be considered the earliest review. The complete description of the specifics of such models was considered in [35]. As far as we know, the control problem for a heterogeneous population was first posed in [36–38], which can also be considered relevant to our work. The difference between the proposed model and models as in [35] is the hierarchical nature of the spread of infection. Initially, the infection exists only at the first, lower level of the hierarchy and subsequently spreads among the population at all levels. The motivation for this spatial model was the idea of studying the epidemic process at the initial stage. According to the Pandemic Intervals Framework (PIF) proposed by the Centers for Disease Control and Prevention [39], the process of infection spread can be divided into six stages (Figure 1). The proposed model is designed to take a closer look at the third interval, which retains the ability to stop the spread of infection at the local level, preventing further spread.

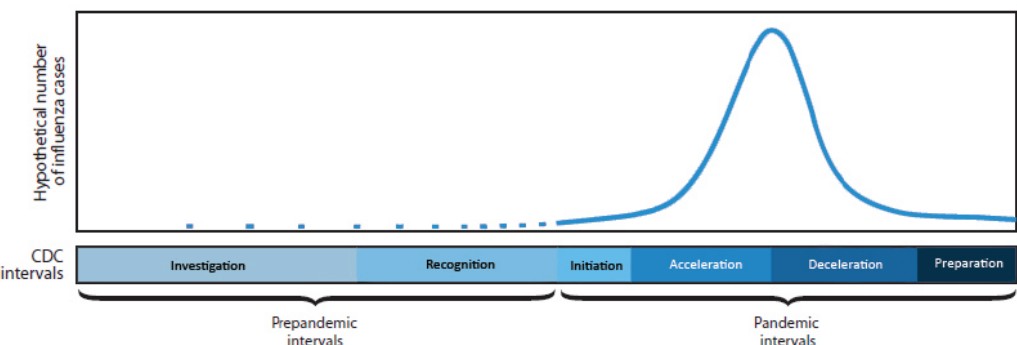

**Figure 1.** CDC pandemic intervals.

In this paper, we establish a hierarchical control-theoretic model to design disease control strategies through quarantine measures to mitigate the impact of epidemics on society. This approach has been motivated by the initial stage of the COVID-19 propagation in China and Italy. The epidemic growth started in a small local village from single cases; then, it propagated to the nearest regions and the whole country. Following this idea, in the considered model and the standard formulation of an epidemic model, the whole population is split into three clusters (levels). Additionally, it is also assumed that migration between the levels is permitted. The presence of migration leads to the epidemic that starts in cluster one, taking over all others. Depending on the migration and infection rates, specific restrictions can be applied to protect the population from epidemics. In analyzing different scenarios of protective measures, the control model is formulated subject to several types of control strategies, such as quarantine in the initial cluster and treatment for all clusters. This approach allows one to estimate the effectiveness of quarantine as a protection

measure for the entire population and to compare a trade-off solution between quarantine and treatment costs. The recent spread of SARS-CoV-19 has occurred due to the wide range of connections between people in modern society in metropolises and large cities.

In the initial period of the COVID-19 pandemic, strict lockdowns and personal restrictions were the only effective measures to decelerate the speed of virus propagation throughout the population. At the same time, the propagation of information about the risk of infection and the presentation of adequate protection measures leads to a decrease in the number of infected residents in the future. In the current study, we model the spread of the virus through susceptible-exposed-infected-recovered (SEIR) dynamics, in which the population is grouped into several subpopulations, i.e., the susceptible (S), the exposed (E), the infected (I), and recovered (R) people on each level. The SEIR dynamics describe the population size evolution, which can be controlled using special quarantine measures and recovery. Recent studies [40–42] have shown the importance of including the exposed subgroup in the model because of the numbers of asymptomatic carriers of the virus who were able to infect other residents of the population.

The objective of this work is to study the epidemics distributed over several clusters and analyze the efficiency of the lockdown on the first level to stop the propagation of the virus among the entire population. We formulate a generalized hierarchical SEIR model, which combines the SEIR model of each level. In the paper, we present a controlled hierarchical SEIR model and show the structure of the optimal policies of quarantine measures on the first level and optimal treatment on all levels of hierarchy. The series of the numerical simulation was run to corroborate the theoretical statements.

The rest of the paper is organized as follows. Section 2 presents the controlled hierarchical SEIR mathematical model. Section 4 describes the optimal control problem. Section 5 presents the series of numerical experiments. Section 6 concludes the paper.

## 2. Deterministic Epidemic Model

In contrast to classical SIRS models [43,44], where populations are divided into three groups, in this section, we formulate a three-level modified SEIR model (Susceptible-Infected-Recovered-Susceptible) with one virus circulated in a population of size $N$. This auxiliary partitioning allows us to capture three epidemic processes that occur in different population clusters. Individual regions, villages, cities within a country, and different countries can be considered clusters. According to this assumption, we can say that if the epidemic has been initiated in the small village (first cluster), then it might be extended to the city and region or country. Formally, this epidemic situation can be formulated as the framework. The virus propagation starts at the first level of hierarchy and continues on the $i$-th levels, $i = 2, 3$ according to the migration rates. We consider this spreading process as the first level hierarchy in the susceptible-quarantined-exposed-infected-recovered-susceptible (SQEIR) model. At the first level, we consider five subgroups: *susceptible* $(S)$, *quarantined* $(Q)$, *exposed* $(E)$, *infected* $(I)$, and *recovered* $(R)$. We suppose that susceptible people are subject to the threat of the virus because recovered people are assumed to have developed immunity, and quarantined people are those who are isolated and therefore cannot be in touch with other people who are potentially infected (in the following, the susceptible proportion of people will be our control variable on the first level). The pathogen infects the exposed and the infected groups, and their contact with other people may infect the susceptible group, but neither the recovered nor the quarantined people. It is also assumed that a subgroup of the exposed group represents residents of the population who do not develop the disease symptoms. The division of the entire population differs from the first cluster in the second and the third clusters, where we have the following subgroups *susceptible* $(S)$, *exposed* $(E)$, *infected* $(I)$, and *recovered* $(R)$. In all clusters, members of the susceptible group become members of the exposed group at the rate of $\beta_i$, $i = \overline{1,3}$; exposed people develop symptoms and become infected people at the rate $k_i$, $i = \overline{1,3}$. For all clusters the self-recovery rates are defined as $\sigma_i$, $i = \overline{1,3}$. The parameter $\delta_1$ shows the rate at which susceptible people choose isolation and become quarantined. At the rate $p_1$, residents

from the quarantine group return to the susceptible group. Moreover, it is supposed that migration occurs between clusters and that migration rates are $m_i^l$, $i = \overline{1,3}$, and $l = 1, 2$, respectively. In the course of time, the protection level of recovered residents vanishes at the rate of $\gamma_i$, $i = \overline{1,3}$. The scheme of the transition is represented in Figure 2.

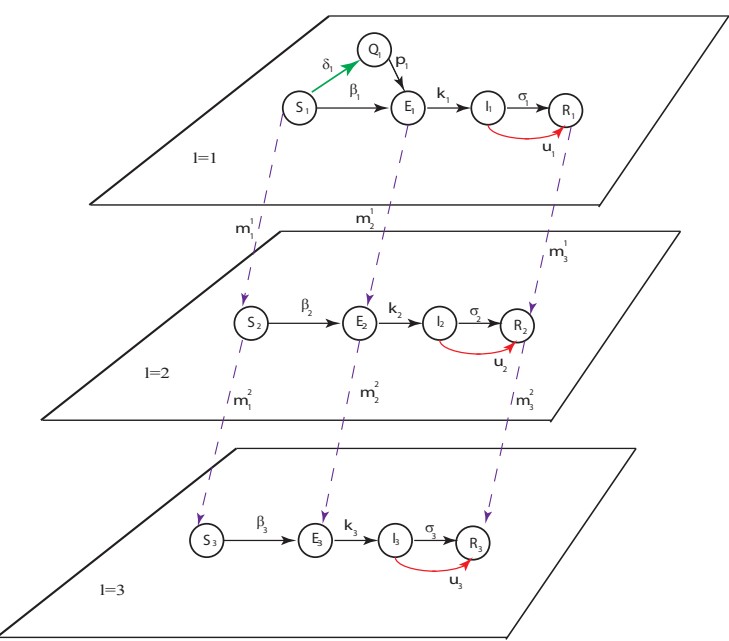

**Figure 2.** The scheme of transition between groups *S, Q, E, I, R*, and hierarchical levels.

We model the epidemic process as a system of nonlinear differential equations, where $N_i(t)$ is size of the population on the level $i$ at the time moment $t$. $n_{S_i}(t)$ indicates the number of susceptible individuals, $n_{Q_1}(t)$ the number of quarantined individuals, $n_{E_i}(t)$ the number of exposed individuals, $n_{I_i}(t)$ the number of infected individuals, and $n_{R_i}(t)$ the number of recovered individuals. The next conditions should be satisfied: $n_{S_1}(t) + n_{Q_1}(t) + n_{E_1}(t) + n_{I_1}(t) + n_{R_1}(t) = N_1(t)$, $n_{S_j}(t) + n_{E_j}(t) + n_{I_j}(t) + n_{R_j}(t) = N_j(t)$ for $j = \overline{2,3}$.

According to previous definitions, the variables $S_i(t), Q_1(t), E_i(t), I_i(t), R_i(t)$ represent the proportions of susceptible, quarantined, exposed, infected, and recovered individuals at time moment $t \in [0, T]$, where

$$S_i(t) = \frac{n_{S_i}(t)}{N_i}, Q_1(t) = \frac{n_{Q_1}(t)}{N_1}, E_i(t) = \frac{n_{E_i}(t)}{N_i}, I_i(t) = \frac{n_{I_i}(t)}{N_i}, R_i(t) = \frac{n_{R_i}(t)}{N_i}.$$

At the beginning of the epidemic, at time $t = 0$, the majority of individuals belong to the susceptible state, and a small fraction of individuals are infected. For all levels the initial states are defined as:

1. $S_1(0) = S^0 \in (0,1)$, $E_1(0) = E_1^0 \in (0,1)$, $I_1(0) = I_1^0 \in (0,1)$, $Q_1(0) = R_1(0) = 0$,
2. $S_2(0) = 1$, $E_2(0) = I_2(0) = R_2(0) = 0$,
3. $S_3(0) = 1$, $E_3(0) = I_3(0) = R_3(0) = 0$.

Systems of nonlinear differential equations represent the propagation of the virus in each cluster/level of the population.

On the first level:

$$\begin{aligned} \dot{S}_1 &= -\beta_1(E_1 + I_1)S_1 - \delta_1 S_1 - m_1^1 S_1; \\ \dot{Q}_1 &= \delta_1 S_1 - p_1 Q_1; \\ \dot{E}_1 &= \beta_1(E_1 + I_1)S_1 + p_1 Q_1 - k_1 E_1 - m_2^1 E_1; \\ \dot{I}_1 &= k_1 E_1 - (\sigma_1 + u_1)I_1; \\ \dot{R}_1 &= (\sigma_1 + u_1)I_1 - m_3^1 R_1. \end{aligned} \tag{1}$$

On the second level:

$$
\begin{aligned}
\dot{S_2} &= -\beta_2(E_2 + I_2)S_2 + m_1^1 S_1 - m_1^2 S_2; \\
\dot{E_2} &= \beta_2(E_2 + I_2)S_2 - k_2 E_2 + m_2^1 E_1 - m_2^2 E_2; \\
\dot{I_2} &= k_2 E_2 - (\sigma_2 + u_2)I_1; \\
\dot{R_2} &= (\sigma_2 + u_2)I_2 + m_3^1 R_1 - m_3^2 R_2.
\end{aligned}
\tag{2}
$$

On the third level:

$$
\begin{aligned}
\dot{S_3} &= -\beta_3(E_3 + I_3)S_3 + m_1^2 S_2; \\
\dot{E_3} &= \beta_3(E_3 + I_3)S_3 - k_3 E_3 + m_2^2 E_2; \\
\dot{I_3} &= k_3 E_3 - (\sigma_3 + u_3)I_3; \\
\dot{R_3} &= (\sigma_3 + u_3)I_3 + m_3^2 R_2.
\end{aligned}
\tag{3}
$$

## 3. Basic Reproduction Number $R_0$

In this section, we use the next generation method (NGM) [45–47] to estimate the basic reproduction number $R_0$ in the model (1)–(3). This parameter shows the average number of cases of an infectious disease arising via transmission from a single infected individual and evaluates the asymptotic behavior of an epidemic process. It is known that the number of infected agents asymptotically decreases if $R_0 > 1$, and increases otherwise. According to the NGM method, the original system is transformed as $\frac{dx_i}{dt} = F_i - V_i$, where its matrix is $F = \frac{dF_i}{dx_i}$, $V = \frac{dV_i}{dx_i}$. Here, $i$ defines indexes of all infected state variables in the initial system of differential equations.

$$
F = \begin{bmatrix}
\beta_1 & \beta_1 & 0 & 0 & 0 & 0 \\
0 & 0 & 0 & 0 & 0 & 0 \\
0 & 0 & \beta_2 & \beta_2 & 0 & 0 \\
0 & 0 & 0 & 0 & 0 & 0 \\
0 & 0 & 0 & 0 & \beta_3 & \beta_3 \\
0 & 0 & 0 & 0 & 0 & 0
\end{bmatrix}; \quad
V = \begin{bmatrix}
k_1 + m_2^1 & 0 & 0 & 0 & 0 & 0 \\
-k_1 & \sigma_1 - u_1 & 0 & 0 & 0 & 0 \\
-m_2^1 & 0 & k_2 + m_2^2 & 0 & 0 & 0 \\
0 & 0 & -k_2 & \sigma_2 - u_2 & 0 & 0 \\
0 & 0 & -m_2^2 & 0 & k_3 & 0 \\
0 & 0 & 0 & 0 & -k_3 & \sigma_1 - u_1
\end{bmatrix};
$$

Eigenvalues of matrix $FV^{-1}$, which defines a non-negative vector giving the expected number of new infections:

$$
\left(0, 0, 0, \frac{\beta_1(k_1 + \sigma_1 + u_1)}{(k_1 + m_2^1)(\sigma_1 + u_1)}, \frac{\beta_2(k_2 + \sigma_2 + u_2)}{(k_2 + m_2^2)(\sigma_2 + u_2)}, \frac{\beta_3(k_3 + \sigma_3 + u_3)}{k_3(\sigma_3 + u_3)}\right)
$$

The method guarantees that a maximum of these values is $R_0$.

Estimating the different combinations of the parameters, we realized the importance of some mixtures of control parameters $u_1(t), u_2(t), u_3(t)$. For example, if the values of parameters are fixed, and if the control rates $u_1(t), u_2(t), u_3(t)$ are constant and equal 0.3, then the coefficient $R_0 = 0.857 < 1$. However, using the same set of parameters and if the controls $u_1(t), u_2(t), u_3(t)$ equal zero, then coefficient $R_0 = 4.2857 > 1$. Figure 3 shows regions of $R_0 < 1$ for different combinations of $\beta_i = \beta, u_i = u$, and $k_i = k$ parameters, $i = \overline{1,3}$.

According to the left diagram in the Figure 3, if the control $u_{max}$ is 1 for the given example, then the parameter $\beta > 0.5$ will cause epidemics to spread with $R_0 > 1$, but if $\beta < 0.4$, which is rather small, then there is always a control $u(t)$ that eliminates epidemics. The figure also confirms the intuitive reasoning that the larger the $\beta$, the more effort is required to stop epidemics.

The diagram on the right in Figure 3 shows that for the given example, the larger the $k$, the fewer efforts are required to eliminate epidemics, and for large enough $k$ epidemics will stop unless the control is too small.

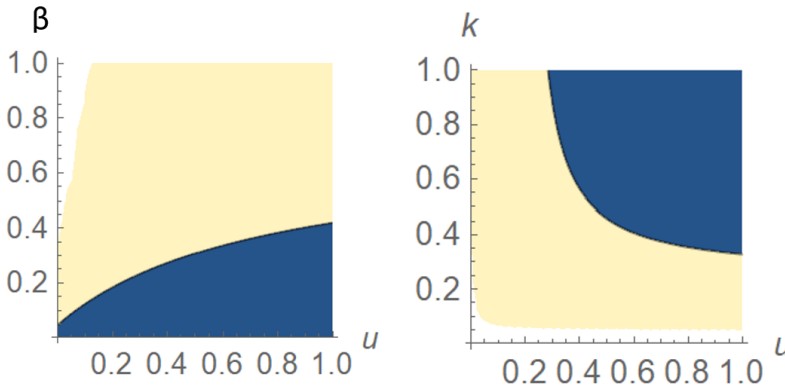

**Figure 3.** Investigation of $R_0(\beta, u)$ and $R_0(k, u)$ for $u_i(t) = u, \beta_i = \beta, k_i = k$. $R_0 < 1$ only in blue regions.

### 4. Optimal Control Problem

The main issue of the current study is the formalization of the optimal control problem to minimize the damage from virus attacks and to improve the protection of the population. In the current model, it is supposed that quarantine and treatment can protect the population from the spread of the virus. At all levels, treatment is used, represented as control strategies $u_1(t)$, $u_2(t)$, $u_3(t)$ in the subpopulations of the infected nodes ($I \to R$), in the case in which the epidemic spreads over all levels of the hierarchy according to the migration rates. Following [33,48,49] we can determine the cost of infection, which is a direct loss of an infected person, such as personal treatment costs, a loss of productivity, or inability to work. The external costs, which are the value that the government pays to allow the health system to treat the infected and increase the probability of recovery for the treated individuals, can be defined as the treatment cost.

**Cost functions.** At any given $t$, functions $f_i(I_i(t))$ represent infection costs, and functions $h_i(u_i(t))$, $i = \overline{1,3}$ correspond to treatment costs. The next conditions should be satisfied for the defined functions. Here functions $f_i(I_i) = \overline{f}_i I_i(t)$ are non-decreasing and twice-differentiable convex functions, $f_i(0) = 0$, $f_i(I_i) > 0$ for $I_i > 0$, $i = \overline{1,3}$, and $h_i(u_i(t)) = \overline{h}_i u_i^2(t)$ is a twice-differentiable and increasing function in $u_i(t)$ such as $h_i(0) = 0$, $h_i(u_i(t)) > 0$, $i = \overline{1,3}$, when $u_i(t) > 0$.

The aggregated system costs on the time interval $[0, T]$ are defined as the functional

$$J = J_1 + J_2 + J_3,$$

where

$$
\begin{aligned}
J_1 &= \int_0^T f_1(I_1(t)) + h_1(u_1(t))dt, \\
J_2 &= \int_0^T f_2(I_2(t)) + h_2(u_2(t))dt, \\
J_3 &= \int_0^T f_3(I_3(t)) + h_3(u_3(t))dt.
\end{aligned}
\tag{4}
$$

The optimal control problem is to minimize these costs, i.e.,

$$\min_{u_1, u_2, u_3} J.$$

Using Pontryagin's maximum principle [50,51] we construct Hamiltonian and adjoint functions as follows. Here, the generalized Hamiltonian of the system is $H = H_1 + H_2 + H_3$. Hamiltonian $H_1$ of the first level:

$$
\begin{aligned}
H_1 = {}& -f_1(I_1) - h_1(u_1) + (\lambda_{E_1} - \lambda_{S_1})\beta_1(E_1 + I_1)S_1 + (\lambda_{Q_1} - \lambda_{S_1})\delta_1 S_1 + \\
& (\lambda_{E_1} - \lambda_{Q_1})p_1 Q_1 + (\lambda_{I_1} - \lambda_{E_1})k_1 E_1 + (\lambda_{R_1} - \lambda_{I_1})(\sigma_1 + u_1)I_1 + \\
& (\lambda_{S_2} - \lambda_{S_1})m_1^1 S_1 + (\lambda_{E_2} - \lambda_{E_1})m_2^1 E_1 + (\lambda_{R_2} - \lambda_{R_1})m_3^1 R_1.
\end{aligned}
\tag{5}
$$

Hamiltonian $H_2$ of the second level:

$$
\begin{aligned}
H_2 = &-f_2(I_2) - h_2(u_2) + (\lambda_{E_2} - \lambda_{S_2})\beta_2(E_2 + I_2)S_2 + (\lambda_{I_2} - \lambda_{E_2})k_2E_2 + \\
&(\lambda_{R_2} - \lambda_{I_2})(\sigma_2 + u_2)I_2 + (\lambda_{S_3} - \lambda_{S_2})m_1^2 S_2 + (\lambda_{E_3} - \lambda_{E_2})m_2^2 E_2 + \\
&(\lambda_{R_3} - \lambda_{R_2})m_3^2 R_2.
\end{aligned}
\tag{6}
$$

Hamiltonian $H_3$ of the third level:

$$
\begin{aligned}
H_3 = &-f_3(I_3) - h_3(u_3) + (\lambda_{E_3} - \lambda_{S_3})\beta_3(E_3 + I_3)S_3 + (\lambda_{I_3} - \lambda_{E_3})k_3E_3 + \\
&(\lambda_{R_3} - \lambda_{I_3})(\sigma_3 + u_3)I_3.
\end{aligned}
\tag{7}
$$

Adjoint functions $\lambda_{S_1}(t), \lambda_{Q_1}(t), \lambda_{E_1}(t), \lambda_{I_1}(t)$, and $\lambda_{R_1}(t)$ of the first level are defined as follows:

$$
\begin{aligned}
\dot{\lambda}_{S_1}(t) &= (\lambda_{S_1} - \lambda_{E_1})\beta_1(E_1 + I_1) + (\lambda_{S_1} - \lambda_{Q_1})\delta_1 + (\lambda_{S_1} - \lambda_{S_2})m_1^1; \\
\dot{\lambda}_{Q_1}(t) &= (\lambda_{Q_1} - \lambda_{E_1})p_1; \\
\dot{\lambda}_{E_1}(t) &= (\lambda_{E_1} - \lambda_{I_1})k_1 + (\lambda_{S_1} - \lambda_{E_1})\beta_1 S_1 + (\lambda_{E_1} - \lambda_{E_2})m_2^1; \\
\dot{\lambda}_{I_1}(t) &= f_1'(I_1) + (\lambda_{S_1} - \lambda_{E_1})\beta_1 S_1 + (\lambda_{I_1} - \lambda_{R_1})(\sigma_1 + u_1); \\
\dot{\lambda}_{R_1}(t) &= (\lambda_{R_1} - \lambda_{R_2})m_3^1,
\end{aligned}
\tag{8}
$$

with the transversality conditions given by

$$
\lambda_{S_1}(T) = \lambda_{Q_1}(T) = \lambda_{E_1}(T) = \lambda_{I_1}(T) = \lambda_{R_1}(T) = 0.
\tag{9}
$$

Analogously, adjoint functions $\lambda_{S_2}(t), \lambda_{E_2}(t), \lambda_{I_2}(t)$, and $\lambda_{R_2}(t)$ of the second level are:

$$
\begin{aligned}
\dot{\lambda}_{S_2}(t) &= (\lambda_{S_2} - \lambda_{E_2})\beta_2(E_2 + I_2) + (\lambda_{S_2} - \lambda_{S_3})m_1^2; \\
\dot{\lambda}_{E_2}(t) &= (\lambda_{E_2} - \lambda_{I_2})k_2 + (\lambda_{S_2} - \lambda_{E_2})\beta_2 S_2 + (\lambda_{E_2} - \lambda_{E_3})m_2^2; \\
\dot{\lambda}_{I_2}(t) &= f_2'(I_2) + (\lambda_{S_2} - \lambda_{E_2})\beta_2 S_2 + (\lambda_{I_2} - \lambda_{R_2})(\sigma_2 + u_2); \\
\dot{\lambda}_{R_2}(t) &= (\lambda_{R_2} - \lambda_{R_3})m_3^2,
\end{aligned}
\tag{10}
$$

with the transversality conditions given by

$$
\lambda_{S_2}(T) = \lambda_{E_2}(T) = \lambda_{I_2}(T) = \lambda_{R_2}(T) = 0.
\tag{11}
$$

Adjoint functions $\lambda_{S_3}(t), \lambda_{E_3}(t), \lambda_{I_3}(t)$, and $\lambda_{R_3}(t)$ of the third level:

$$
\begin{aligned}
\dot{\lambda}_{S_3}(t) &= (\lambda_{S_3} - \lambda_{E_3})\beta_3(E_3 + I_3); \\
\dot{\lambda}_{E_3}(t) &= (\lambda_{E_3} - \lambda_{I_3})k_3 + (\lambda_{S_3} - \lambda_{E_3})\beta_3 S_3; \\
\dot{\lambda}_{I_3}(t) &= f_3'(I_3) + (\lambda_{S_3} - \lambda_{E_3})\beta_3 S_3 + (\lambda_{I_3} - \lambda_{R_3})(\sigma_3 + u_3); \\
\dot{\lambda}_{R_3}(t) &= 0,
\end{aligned}
\tag{12}
$$

with the transversality conditions given by

$$
\lambda_{S_3}(T) = \lambda_{E_3}(T) = \lambda_{I_3}(T) = \lambda_{R_3}(T) = 0.
\tag{13}
$$

According to Pontryagin's maximum principle, there exist continuous and piece-wise continuously differentiable co-state functions

$$
\lambda_r(t), \; r \in \{S_1, Q_1, E_1, \ldots, I_3, R_3\}
$$

that satisfy (8)–(13) at every time $t \in [0, T]$, together with continuous functions $u_1^*(t), u_2^*(t)$, and $u_3^*(t)$:

$$
(u_1^*, u_2^*, u_3^*) \in \arg \max_{u_1, u_2, u_3 \in [0, u_{max}]} H(\lambda, S_1, Q_1, \ldots, R_3, u_1, u_2, u_3).
\tag{14}
$$

Functions $\varphi_i(t)$, $i = \overline{1,3}$ are defined as follows:

$$
\begin{aligned}
\varphi_1(t) &= (\lambda_{R_1}(t) - \lambda_{I_1}(t))I_1(t), \\
\varphi_2(t) &= (\lambda_{R_2}(t) - \lambda_{I_2}(t))I_2(t), \\
\varphi_3(t) &= (\lambda_{R_3}(t) - \lambda_{I_3}(t))I_3(t).
\end{aligned}
\tag{15}
$$

Following Pontryagin's maximum principle, we construct the optimal treatment strategies.

**Proposition 1.** *When $h_i(\cdot)$ are concave functions, then the optimal control structure takes the following form for any $i = \overline{1,3}$:*

$$
u_i^*(t) = \begin{cases} 0, & \text{for } \varphi_i(t) < h_i(u_{max}); \\ u_{max}, & \text{for } \varphi_i(t) \geq h_i(u_{max}). \end{cases}
\tag{16}
$$

**Proposition 2.** *When $h_i(\cdot)$ are strictly convex functions, then the optimal control structure takes the following form for any $i = \overline{1,3}$:*

$$
u_i^*(t) = \begin{cases} 0, & \text{for } \varphi_i(t) \leq h_i'(0); \\ h'^{-1}(\varphi_i), & \text{for } h_i'(0) < \varphi_i(t) \leq h_i'(u_{max}); \\ u_{max}, & \text{for } h_i'(u_{max}) < \varphi_i(t). \end{cases}
\tag{17}
$$

*where value $u_{max} \in [0,1]$.*

To prove this propositions, we can rewrite the Hamiltonian in terms of the function $\varphi_i(t)$. After that, we obtain:

$$
\begin{aligned}
H = {} & -f_1(I_1) - f_2(I_2) - f_3(I_3) + (\lambda_{E_1} - \lambda_{S_1})\beta_1(E_1 + I_1)S_1 + (\lambda_{Q_1} - \lambda_{S_1})\delta_1 S_1 + \\
& (\lambda_{E_1} - \lambda_{Q_1})p_1 Q_1 + (\lambda_{I_1} - \lambda_{E_1})k_1 E_1 + (\lambda_{R_1} - \lambda_{I_1})\sigma_1 I_1 + (\lambda_{S_2} - \lambda_{S_1})m_1^1 S_1 + \\
& (\lambda_{E_2} - \lambda_{E_1})m_2^1 E_1 + (\lambda_{R_2} - \lambda_{R_1})m_3^1 R_1 + (\lambda_{E_2} - \lambda_{S_2})\beta_2(E_2 + I_2)S_2 + \\
& (\lambda_{I_2} - \lambda_{E_2})k_2 E_2 + (\lambda_{R_2} - \lambda_{I_2})\sigma_2 I_2 + (\lambda_{S_3} - \lambda_{S_2})m_1^2 S_2 + (\lambda_{E_3} - \lambda_{E_2})m_2^2 E_2 + \\
& (\lambda_{R_3} - \lambda_{R_2})m_3^2 R_2 + (\lambda_{E_3} - \lambda_{S_3})\beta_3(E_3 + I_3)S_3 + (\lambda_{I_3} - \lambda_{E_3})k_3 E_3 + \\
& (\lambda_{R_3} - \lambda_{I_3})\sigma_3 I_3 + (\varphi_1 u_1 - h_1(u_1)) + (\varphi_2 u_2 - h_2(u_2)) + (\varphi_3 u_3 - h_3(u_3)).
\end{aligned}
\tag{18}
$$

We can divide this maximization problem into three subproblems and find the optimal controls $u_1^*(t)$, $u_2^*(t)$, and $u_3^*(t)$, separately:

$$
\max_{u_1}[-h_1(u_1) + \varphi_1 u_1] + \max_{u_2}[-h_2(u_2) + \varphi_2 u_2] + \max_{u_3}[-h_3(u_3) + \varphi_3 u_3].
\tag{19}
$$

For any admissible control $\overline{u_i}$ and according to (18) for all $t \in [0, T]$ we arrive at

$$
-h_i(u_i) + u_i \varphi_i \geq -h_i(\overline{u_i}) + \overline{u_i}\varphi_i.
\tag{20}
$$

Since $\overline{u_i} = 0$ is an admissible control, we obtain

$$
-h_i(u_i) + u_i \varphi_i \geq -h_i(0) + 0 \cdot \varphi_i = 0.
\tag{21}
$$

According to the algorithm of Pontryagin's maximum principle to determine the optimal control structure, we consider the next derivatives:

$$
\frac{\partial H}{\partial u_i} = -\dot{h}_i(u_i) + \varphi_i = 0, \ \ i = \overline{1,3}.
\tag{22}
$$

As $h_i(\cdot)$ are increasing functions and $I_i(\cdot) \geq 0$, then the Hamiltonian reaches its maximum if $\varphi_i(t) = \dot{h}_i(u_i(t)) \geq 0$, $i = 1, 2, 3$. According to (15), since $I_i(t) \geq 0$ for all $t \in [0, T]$, this condition is satisfied if and only if $\lambda_{R_i}(t) - \lambda_{I_i}(t) \geq 0$ for $i = \overline{1,3}$. To complete the proof of proposition, we consider the auxiliary lemma.

**Lemma 1.** *For all $t \in [0, T]$, we have $\lambda_{R_i}(t) - \lambda_{I_i}(t) \geq 0$, where $i = \overline{1, 3}$.*

The Proof of Lemma 1 consists of two parts: firstly, we consider the case when $t = T$ and show that derivatives of the functions $\lambda_{R_i}(t) - \lambda_{I_i}(t)$, $i = \overline{1, 3}$ are non-positive; secondly, we prove by contradiction that on the whole interval $[0, T]$ these functions are non-negative. The complete proofs of Lemma 1 are presented in Appendix A and follow the same technique as that in [48,52].

### 4.1. Functions $h_i(\cdot)$ Are Concave

Let $h_i(\cdot)$ be a concave function ($h_i''(\cdot) \leq 0$); then, according to (5)–(7) the Hamiltonian is a convex function of $u_i(\cdot)$, $i = \overline{1, 3}$. There are two different options for $u_i \in [0, u_{max}]$ that maximimize the Hamiltonian. Parameter $u_{max} \in [0, 1]$.

If $-h_i(0) + \varphi_i(t) \cdot 0 > -h_i(u_{max}) + \varphi_i(t) \cdot u_{max}$ or $h_i(u_{max}) > \varphi_i(t) u_{max}$, then the optimal control is $u_i = 0$ (see Figure 4 (left)); otherwise $u_i = u_{max}$ (see Figure 4 (right)).

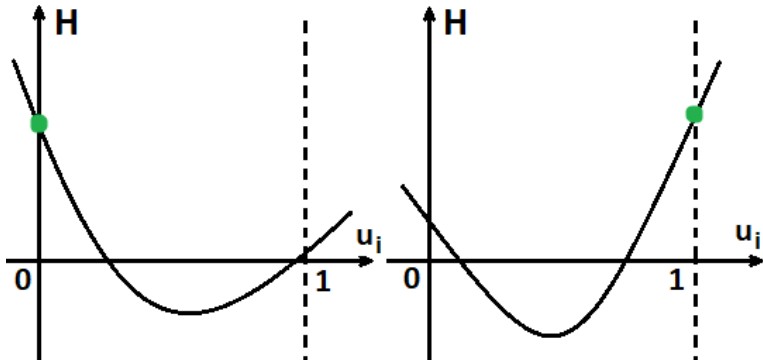

**Figure 4.** Hamiltonian function in a case in which functions $h_i(\cdot)$ are concave.

For $i = \overline{1, 3}$, the optimal control parameters $u_i(t)$ are defined as follows:

$$u_i^*(t) = \begin{cases} 0, & \text{for } \varphi_i(t) u_{max} < h_i(u_{max}), \\ u_{max}, & \text{for } \varphi_i(t) u_{max} \geq h_i(u_{max}). \end{cases} \tag{23}$$

### 4.2. Functions $h_i(\cdot)$ Are Strictly Convex

Let $h_i(\cdot)$ be a strictly convex function ($h_i''(\cdot) > 0$); then, the Hamiltonian is concave function. Consider the following derivative:

$$\frac{\partial}{\partial x}(-h_i(x) + \varphi_i(t)x)\mid_{x=x_i} = 0, \tag{24}$$

where $x \in [0, u_{max}]$, $u_i^*(t) = x_i$, and $u_{max} \in [0, 1]$. There are three different types of points at which the Hamiltonian reaches its maximum (Figure 5). To find them, we need to consider the derivatives of the Hamiltonian at $u_i = 0$ and $u_i = u_{max}$. If the derivatives (24) at $u_i = 0$ are non-increasing ($-h_i'(0) + \varphi_i(t) \leq 0$), then the value of the control that maximizes the Hamiltonian is less than 0, and according to our restrictions ($u_i \in [0, u_{max}]$) the optimal control will be equal to 0 (Figure 5a). If the derivatives at $u_i = u_{max}$ are increasing ($-h_i'(u_{max}) + \varphi_i(t) > 0$), this means that the value of the control that maximizes the Hamiltonian is greater than $u_{max}$. Hence, the optimal control will be 1 (Figure 5c); otherwise, we can find the value of $u_i^* \in (0, u_{max})$ (see Figure 5b):

$$u_i^*(t) = \begin{cases} 0, & \text{for } \varphi_i(t) \leq h_i'(0), \\ h'^{-1}(\varphi_i(t)), & \text{for } h_i'(0) < \varphi_i(t) \leq h_i'(u_{max}), \\ u_{max}, & \text{for } h_i'(u_{max}) < \varphi_i(t), \; i = 1, 2, 3. \end{cases} \tag{25}$$

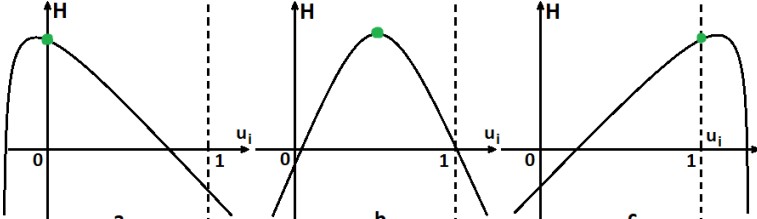

**Figure 5.** Hamiltonian function in a case in which functions $h_i(\cdot)$ are convex.

## 5. Numerical Simulation

In this section, we present several numerical experiments to corroborate our results. For the experiments, we partially use modeled data and real data obtained during the COVID-19 pandemic [33,40,41,53] for the following initial distribution between susceptible, exposed, and infected groups: $S_1^0 = 0.7$, $E_1^0 = 0.1$, $I_1^0 = 0.2$. The spreading and self-recovery rates are defined as $\beta_i = 0.25$, $\sigma_i = 0.1$ for $i = \overline{1,3}$, incubation rates are $k_i = 0.15$ for $i = \overline{1,3}$. The cost functions in infected group are $f_i(I_i(t)) = \overline{f}_i I_i(t)$, and the treatment cost functions are defined as $h_i(u_i(t)) = \overline{h}_i u_i^2(t)$ for $i = \overline{1,3}$. In the current model it is supposed that the maximum value of the control $u_{max} = 0.2$. More detailed explanation of the parameters and initial data used in experiments can be found in Table 1.

**Experiment 1.** In the current experiment, we present the SQEIR model under the assumption that quarantine is not possible for susceptible people. Applying the specially designed procedure and considering the cluster structure of the population, we obtain the result that the epidemic starts and reaches its maximum faster at the first stage than at stages $l = 2, 3$. At the second and the third clusters, the epidemic starts with delays; Figure 6 demonstrates this fact.

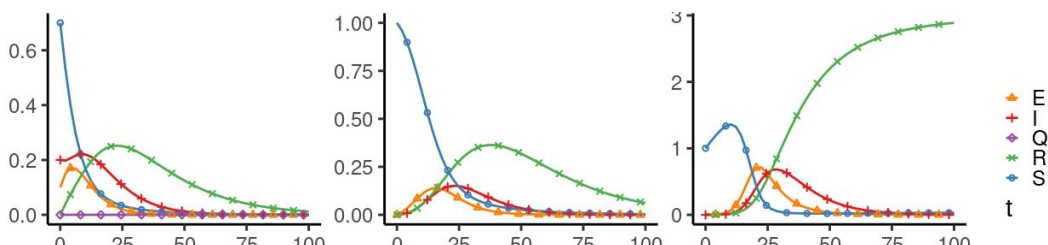

**Figure 6.** Experiment 1. Spread of the virus in three different clusters in an uncontrolled case.

The behaviour of the system in the controlled case is shown in Figure 7.

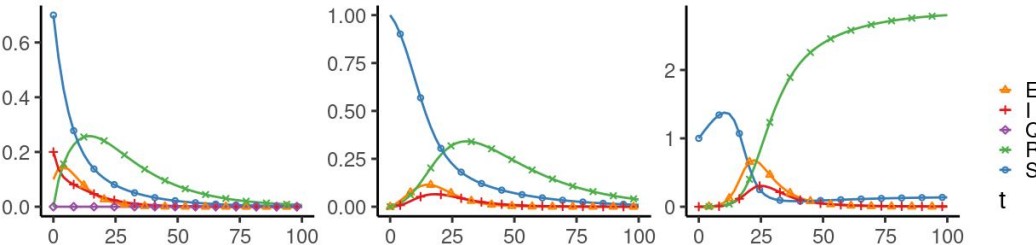

**Figure 7.** Experiment 1. Spread of the virus in three different clusters in a controlled case.

Figures 8 and 9 represents the structure of the optimal treatment policies $u_i(t)$, $i = 1, 2, 3$, and summarizes the changes in all fractions of the population, respectively.

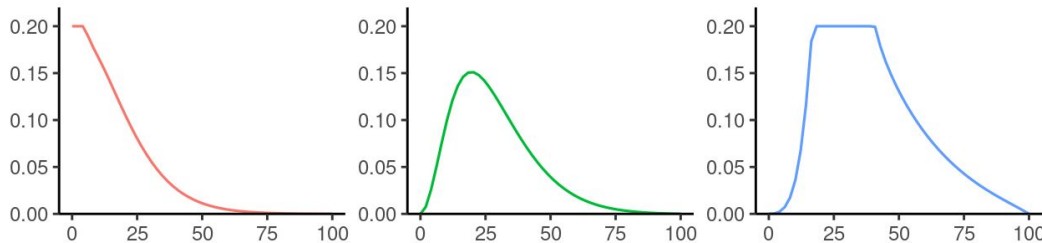

**Figure 8.** Experiment 1. The structure of the optimal treatment policies $u_i(t)$, $i = 1, 2, 3$.

**Table 1.** Parameters used for simulations in experiments.

| Parameters Used for Simulations | | | | | |
|---|---|---|---|---|---|
| **Parameter Name** | **Exp. 1** | **Exp. 2** | **Exp. 3** | **Exp. 4** | **Exp. 5** |
| Fraction of susceptible people at time $t = 0$ ($S_i^0$) | 0.7<br>1<br>1 | 0.7<br>1<br>1 | 0.7<br>1<br>1 | 0.7<br>1<br>1 | 0.7<br>1<br>1 |
| Fraction of exposed people at time $t = 0$ ($E_i^0$) | 0.1<br>0<br>0 | 0.1<br>0<br>0 | 0.1<br>0<br>0 | 0.1<br>0<br>0 | 0.1<br>0<br>0 |
| Fraction of infected people at time $t = 0$ ($I_i^0$) | 0.2<br>0<br>0 | 0.2<br>0<br>0 | 0.2<br>0<br>0 | 0.2<br>0<br>0 | 0.2<br>0<br>0 |
| Infection rate from $S_i$ to $E_i$ ($\beta_i$) | 0.25 | 0.25 | 0.25 | 0.25 | 0.25 |
| Recovery rate ($\sigma_i$) | 0.1 | 0.1 | 0.1 | 0.1 | 0.1 |
| Asymptomatic to Infected ($k_i$) | 0.15 | 0.15 | 0.15 | 0.15 | 0.15 |
| Voluntary self-isolation ($\delta$) | 0 | 0.4 | 0.6 | 0.3 | 0–1 |
| Return from self-isolation ($p$) | 0 | 0.05 | 0 | 0.2 | 0–0.5 |
| Migration rates from $S_i$ to $S_{i+1}(m_1^i)$ | 0.05 | 0.05 | 0.05 | 0.05 | 0–0.3 |
| Migration rates from $E_i$ to $E_{i+1}(m_2^i)$ | 0.05 | 0.05 | 0.05 | 0.05 | 0–0.3 |
| Migration rates from $R_i$ to $R_{i+1}(m_3^i)$ | 0.05 | 0.05 | 0.05 | 0.05 | 0–0.3 |
| Infection costs $\overline{f}_i$ | 1 | 1 | 1 | 1 | 1 |
| Treatment costs $\overline{h}_i$ | 1 | 1 | 1 | 1 | 1 |
| Maximum values of control ($u_{max}$) | 0.2 | 0.2 | 0.2 | 0.2 | 0.2 |
| Aggregated costs $J$ Uncontrolled case | 29.41 | 29.43 | 23.57 | 29.58 | — |
| Aggregated costs $J$ Controlled case | 13.22 | 14.07 | 10.60 | 13.52 | *depend on the parameters* |

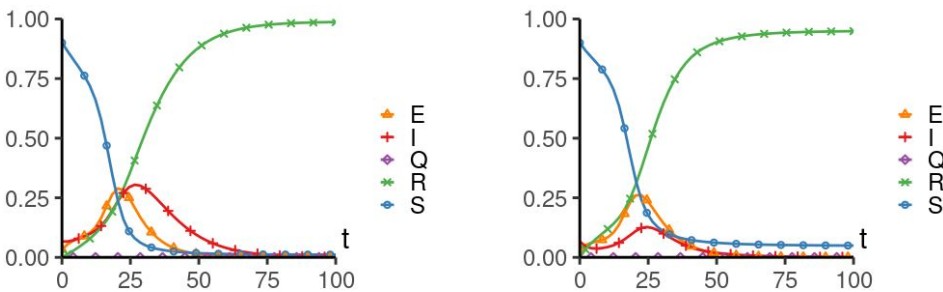

**Figure 9.** Experiment 1. Sum of the fractions of all levels: uncontrolled (**left**) and controlled cases (**right**).

Aggregation system costs in the uncontrolled case are equal to 29.41 monetary units (m.u.) and 13.22 m.u in the controlled case. A comparison of these costs is presented in Figure 10.

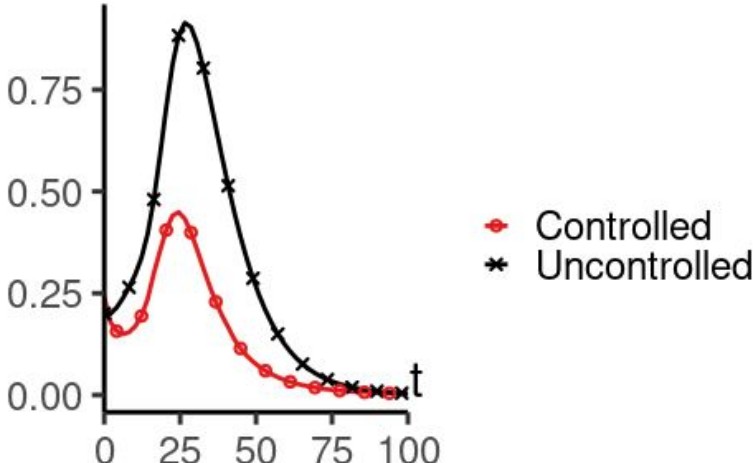

**Figure 10.** Experiment 1. Aggregation costs in uncontrolled and controlled cases.

**Experiment 2.** In the current experiment, we consider the SQEIR model with quarantine ($\delta_1 = 0.4$, $p_1 = 0.05$). We assume that the government has imposed quarantine measures and that citizens should stay at home. This experiment shows the effects of quarantine measures on the development of an epidemic situation. The behavior of the system in the uncontrolled and controlled cases is shown in Figures 11 and 12.

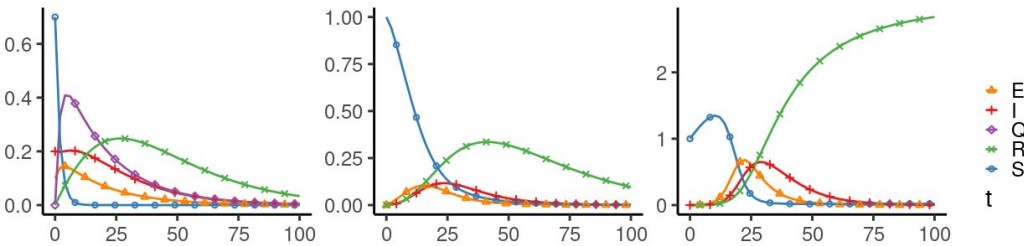

**Figure 11.** Experiment 2. Spread of the virus in three different clusters in the uncontrolled case.

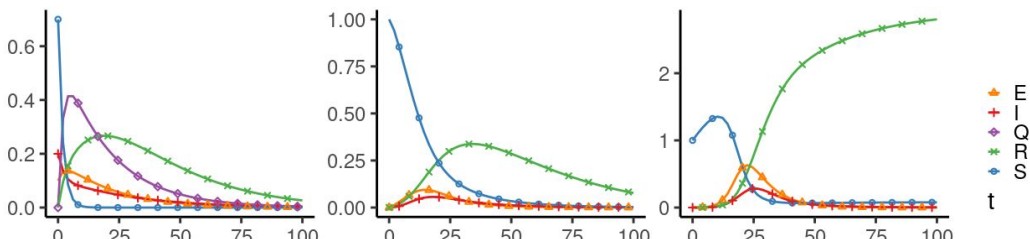

**Figure 12.** Experiment 2. Spread of the virus in three different clusters in the controlled case.

Figure 13 represents the structure of the optimal treatment policies $u_i(t)$, $i = 1, 2, 3$.

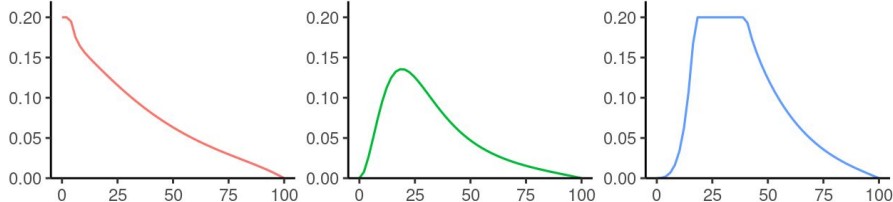

**Figure 13.** Experiment 2. The structure of the optimal treatment policies $u_i(t)$, $i = 1, 2, 3$.

Aggregate changes in all fractions of the population are shown in Figure 14. The aggregated system cost in the uncontrolled case is 29.43 million euros compared with 14.07 million euros in the controlled case (Figure 15).

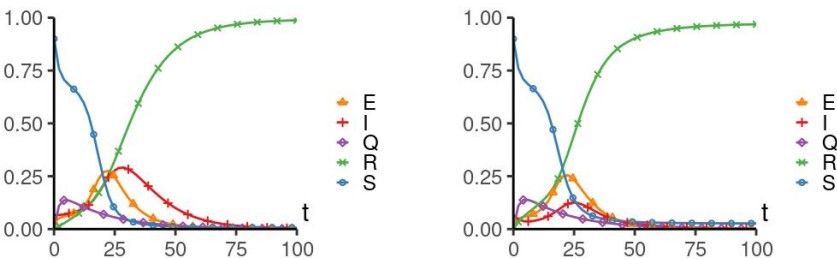

**Figure 14.** Experiment 2. Sum of the fractions of all levels: uncontrolled (**left**) and controlled cases (**right**).

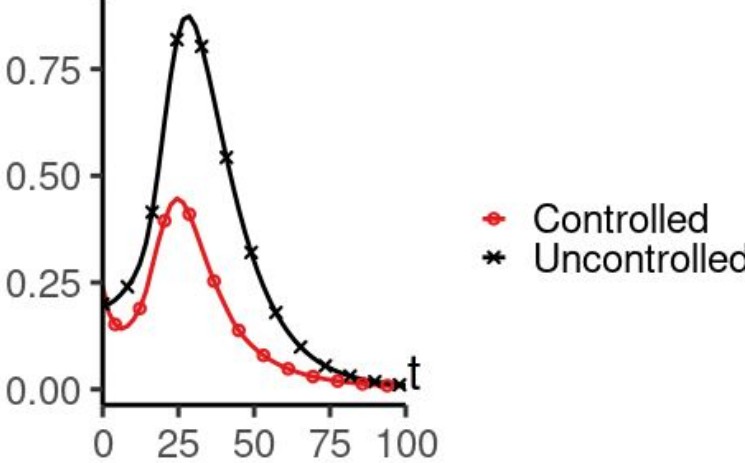

**Figure 15.** Experiment 2. Aggregation costs in uncontrolled and controlled cases.

The presence of a quarantine slightly increases the aggregate cost, but the peak number of infected individuals is much lower. Therefore, we can say that optimal controllers save the health system's resources (medical staff, equipment) and prevent its collapse.

**Experiment 3.** In this experiment, it is assumed that all individuals are responsible. We suppose that resident responsibility positively reduces the overall spread of the epidemic

and minimizes aggregate costs. Residents voluntarily remain in quarantine until the end of the interval under consideration $[0, 100]$. This assumption leads to the transition from the susceptible fraction to quarantine in cluster $l = 1$ ($S_1 \rightarrow Q_1$), which occurs at the rate of $\delta_1 = 0.6$. The value $\delta_1$ is higher than in experiment 1, but unlike the previous case, the transition rate from quarantine to exposed ($Q_1 \rightarrow E_1$) is $p_1 = 0$ (lower than in the previous experiment). The other parameters of the system remain the same.

Figures 16 and 17 represent the behavior of the system in uncontrolled and controlled cases, respectively.

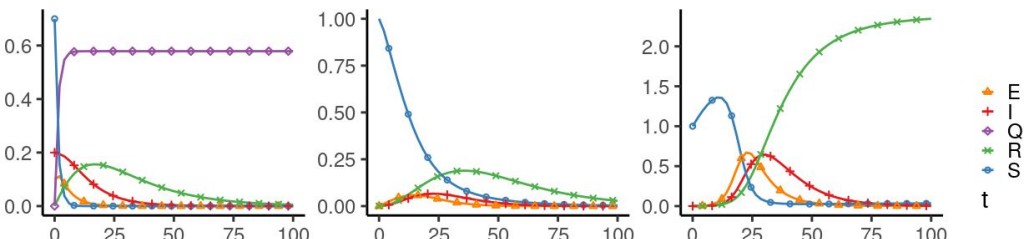

**Figure 16.** Experiment 3. Spread of the virus in the uncontrolled case (responsible individuals). (**Left**): 1 level, (**middle**): 2 level, (**right**): 3 level.

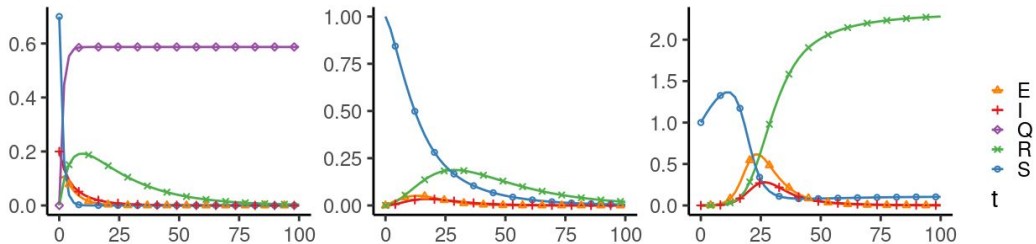

**Figure 17.** Experiment 3. Spread of the virus in the controlled case (responsible individuals). (**Left**): 1 level, (**middle**): 2 level, (**right**): 3 level.

As in the previous experiments, Figure 18 represents the structure of the optimal treatment policy $u_i(t)$, $i = 1, 2, 3$ and provides a summary of changes in all fractions of the population Figure 19. It can be seen that the number of infected people is lower than in the case in which the residents leave quarantine independently without the government's consent.

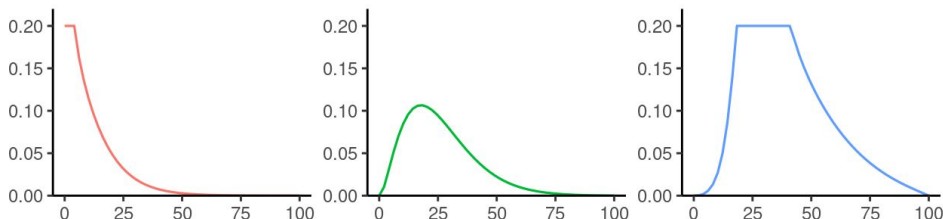

**Figure 18.** Experiment 3. The structure of the optimal treatment policies $u_i(t)$, $i = 1, 2, 3$ (responsible individuals).

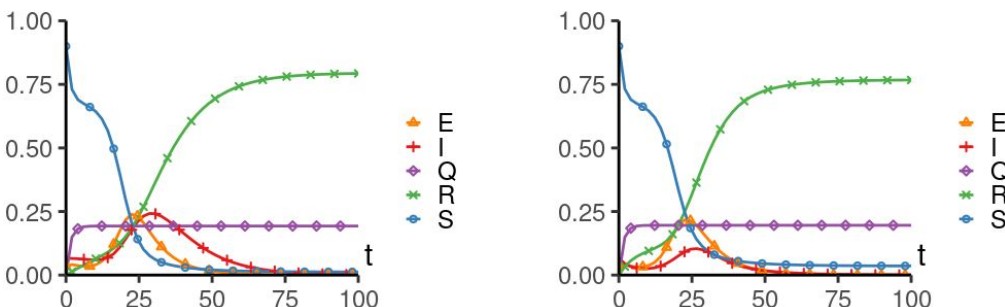

**Figure 19.** Experiment 3. Sum of the fractions of all levels: uncontrolled (**left**) and controlled cases (**right**) with responsible individuals.

Aggregation system costs in the uncontrolled case are equal to 23.57 m.u. and 10.60 m.u in the controlled case (Figure 20).

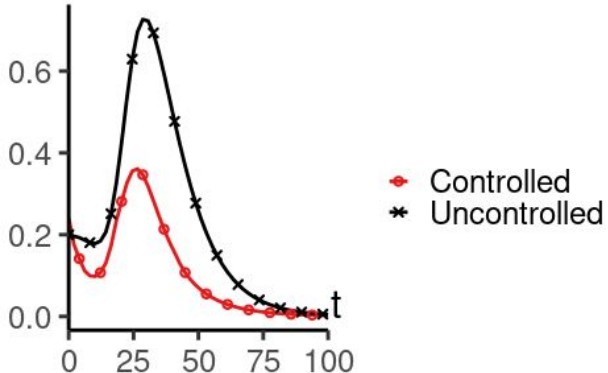

**Figure 20.** Experiment 3. Aggregation costs in uncontrolled and controlled cases with responsible individuals.

**Experiment 4.** Here it is assumed that all individuals in the population are irresponsible and prefer to violate the quarantine isolation rules. Under this assumption, the transition rate from susceptible to quarantined subgroups ($S_1 \rightarrow Q_1$) is chosen as $\delta_1 = 0.3$, which is lower than in experiments 1 and 3. At the same time, quarantined individuals become exposed at a higher rate of $p_1 = 0.2$. All other parameters remain the same. Figures 21 and 22 show the uncontrolled case and the controlled case, respectively.

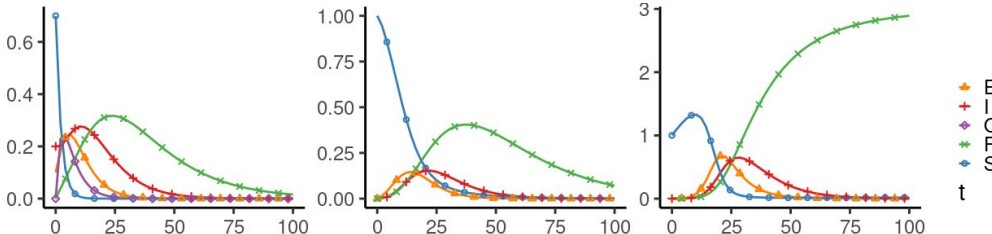

**Figure 21.** Experiment 4. Spread of the virus in the uncontrolled case (irresponsible individuals). (**Left**): 1 level, (**middle**): 2 level, (**right**): 3 level.

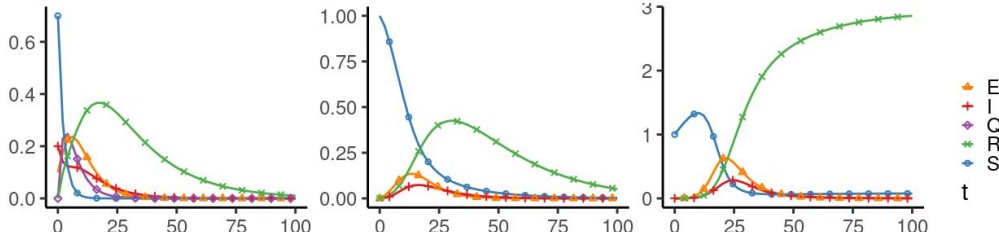

**Figure 22.** Experiment 4. Spread of the virus in the controlled case (irresponsible residents). (**Left**): 1 level, (**middle**): 2 level, (**right**): 3 level.

Figure 23 represents the structure of the optimal treatment policies $u_i(t)$, $i = 1, 2, 3$ and summarizes changes in all fractions of the population Figure 24.

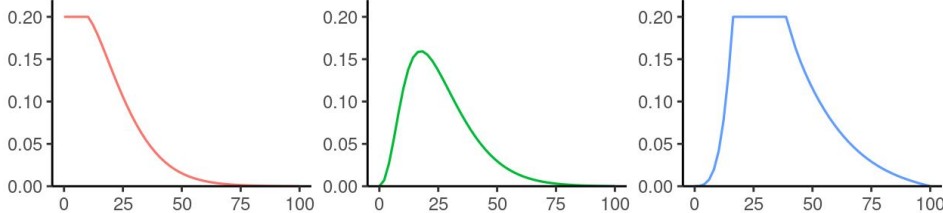

**Figure 23.** Experiment 4. The structure of the optimal treatment policies $u_i(t)$ irresponsible individuals.

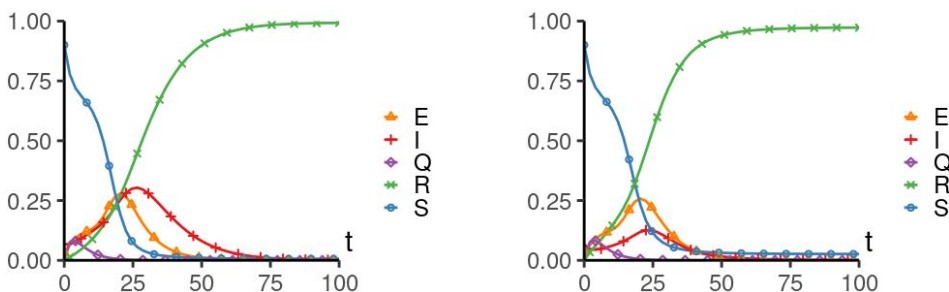

**Figure 24.** Experiment 4. Sum of the fractions of all levels: uncontrolled (**left**) and controlled cases (**right**) with irresponsible individuals.

Aggregate system costs in the uncontrolled case are equal to 29.58 m.u., compared to 13.52 m.u in the controlled case (Figure 25). Comparing the costs with the previous experiment, the hypothesis of the general responsibility of residents in the population is confirmed. The more responsible the residents are in the country, the less the aggregated costs from the epidemic will be.

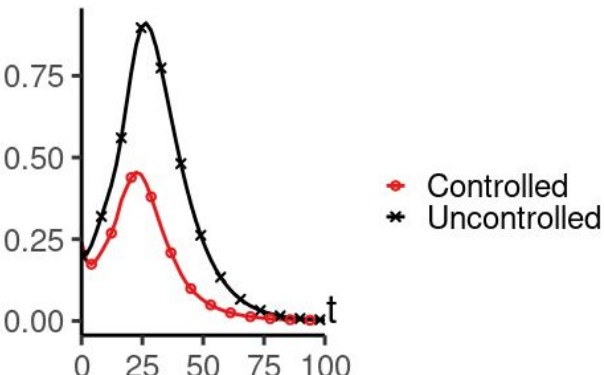

**Figure 25.** Experiment 4. Aggregation costs in uncontrolled and controlled cases with irresponsible individuals.

**Experiment 5.** In this experiment we show how the aggregated costs change in the controlled cases for different parameter values $\delta_1$ and $p_1$ (see Figure 26).

| Aggregated costs in controlled cases | | | | | | |
|---|---|---|---|---|---|---|
| | | $\delta$ | | | | |
| | | **0** | **0.2** | **0.4** | **0.6** | **0.8** | **1** |
| | **0** | 13.22 | 11.25 | 10.80 | 10.60 | 10.47 | 10.39 |
| | **0.1** | 13.22 | 13.78 | 13.78 | 13.78 | 13.78 | 13.78 |
| **p** | **0.2** | 13.22 | 13.53 | 13.52 | 13.51 | 13.50 | 13.49 |
| | **0.3** | 13.22 | 13.45 | 13.42 | 13.40 | 13.40 | 13.39 |
| | **0.4** | 13.22 | 13.41 | 13.37 | 13.36 | 13.35 | 13.34 |
| | **0.5** | 13.22 | 13.40 | 13.35 | 13.33 | 13.33 | 13.32 |

**Figure 26.** Experiment 5. Table of the dependence of aggregated costs on various values of the parameters $\delta_1$ and $p_1$ in the controlled case.

It can be noted that the presence of quarantine measures ($\delta_1 > 0$) brings about lower costs than the absence of these measures ($\delta_1 = 0$) only when people stay in quarantine without leaving it ($p_1 = 0$).

## 6. Conclusions

This paper presents a multilevel modification of the susceptible-quarantine-exposed-infected-recovered (SQEIR) model of the simultaneous spread of a virus over three clusters in the entire population. We have obtained the optimal control structure and the restrictions of feasible controls for a particular class of cost functions. Numerical simulations confirmed the hypothesis that if population residents are compliant with quarantine rules on the first level, then this produces a positive effect, reducing the overall spread of epidemics and the aggregate system costs. We will further investigate specific cases of epidemic spreading in different countries using appropriate statistic datasets. Other research directions include applying the hierarchical SQEIR model to complex networks with different topologies and using vaccination as a control strategy.

**Author Contributions:** E.G., V.T. described the proposed framework and wrote the whole manuscript; I.P., V.T. and D.F. implemented the simulation experiments; E.G., V.T. and I.P. collected data; E.G., V.T. and D.F. revised the manuscript. All authors have read and agreed to the published version of the manuscript.

**Funding:** The research has been partially supported by RFBR according to the research project 20-31-90133.

**Institutional Review Board Statement:** Not applicable.

**Informed Consent Statement:** Not applicable.

**Data Availability Statement:** Data is contained within the article.

**Conflicts of Interest:** The authors declare no conflict of interest.

## Appendix A

**The proof of Lemma 1.** Proof of the Lemma 1 is based on the following properties.

**Property A1.** *Let $v(t)$ be a continuous and piecewise differential function of t. Let $v(t_1) = L$ and $v(t) > L$ for all $t \in (t_1, \ldots, t_0]$. Then $\dot{v}(t_1^+) \geq 0$ where $v(t_1^+) = \lim_{x \to 0} v(x)$.*

**Property A2.** *For any convex and differentiable function $y(x)$, which is 0 at $x = 0$, $y'(x)x - y(x) \geq 0$ for all $x \geq 0$.*

Let us split our proof into two parts. In the first part we will consider the case when $t = T$ and show that derivatives of the functions $(\lambda_{R_i}(t) - \lambda_{I_i}(t))$, $i = \overline{1,3}$ are less than or equal to zero to prove that they are non-increasing at $t = T$. In the second part, we will use the method of proof by contradiction and show that on the whole interval $[0, T]$, these functions are also non-negative.

**Part I.** At time moment $T$, we have according to (9), (11) and (13):

$$\begin{aligned}
(\lambda_{R_1}(T) - \lambda_{I_1}(T)) &= 0; \\
(\lambda_{R_2}(T) - \lambda_{I_2}(T)) &= 0; \\
(\lambda_{R_3}(T) - \lambda_{I_3}(T)) &= 0.
\end{aligned} \tag{A1}$$

From (8), (10) and (12) we obtain

$$\begin{aligned}
(\dot{\lambda}_{R_1}(T) - \dot{\lambda}_{I_1}(T)) &= -\dot{f}_1(I_1(T)) \le 0; \\
(\dot{\lambda}_{R_2}(T) - \dot{\lambda}_{I_2}(T)) &= -\dot{f}_2(I_2(T)) \le 0; \\
(\dot{\lambda}_{R_3}(T) - \dot{\lambda}_{I_3}(T)) &= -\dot{f}_3(I_3(T)) \le 0.
\end{aligned} \tag{A2}$$

Now we can observe that at time moment $T$ all functions are equal to zero and their derivatives are less or equal to zero; then, we observe that $(\lambda_{R_i}(t) - \lambda_{I_i}(t))$, $i = \overline{1,3}$ are decreasing functions at $t = T$.

**Part II.** (Proof by contradiction)

Let $0 \le t^* < T$ be the last instant moment at which one of these inequality constraints for $i = \overline{1,3}$ and $p = \overline{1,2}$ is satisfied:

$$\begin{aligned}
(\lambda_{R_i}(t) - \lambda_{I_i}(t)) \ge 0, && (\lambda_{E_i}(t) - \lambda_{S_i}(t)) \ge 0, && (\lambda_{I_i}(t) - \lambda_{E_i}(t)) \ge 0, \\
(\lambda_{Q_1}(t) - \lambda_{S_1}(t)) \ge 0, && (\lambda_{E_1}(t) - \lambda_{Q_1}(t)) \ge 0, && (\lambda_{S_{i+1}}(t) - \lambda_{S_i}(t)) \ge 0, \\
(\lambda_{E_{i+1}}(t) - \lambda_{E_i}(t)) \ge 0, && (\lambda_{R_{i+1}}(t) - \lambda_{R_i}(t)) \ge 0.
\end{aligned} \tag{A3}$$

For the proof by contradiction we suppose that at the time moment $t^*$ one of the inequalities is equal to zero, i.e., $(\lambda_{R_i}(t^*) - \lambda_{I_i}(t^*)) = 0$, and consider the derivative:

$$\begin{aligned}
\dot{\lambda}_{R_1}(t^*) - \dot{\lambda}_{I_1}(t^*) &= (\lambda_{R_1} - \lambda_{R_2})m_3^1 - f_1'(I_1) - (\lambda_{S_1} - \lambda_{E_1})\beta_1 S_1 - \\
&\quad (\lambda_{I_1} - \lambda_{R_1})(\sigma_1 + u_1).
\end{aligned} \tag{A4}$$

Let us express this function by means of the Hamiltonian

$$\begin{aligned}
\lambda_{R_1}(t) - \lambda_{I_1}(t) = {} & \tfrac{1}{(\sigma_1 + u_1)I_1}[H + f_1(I_1) + f_2(I_2) + f_3(I_3) + h_1(u_1) + h_2(u_2) + \\
& h_3(u_3) - (\lambda_{E_1} - \lambda_{S_1})\beta_1(E_1 + I_1)S_1 - (\lambda_{Q_1} - \lambda_{S_1})\delta_1 S_1 - (\lambda_{E_1} - \lambda_{Q_1})p_1 Q_1 - \\
& (\lambda_{I_1} - \lambda_{E_1})k_1 E_1 - (\lambda_{S_2} - \lambda_{S_1})m_1^1 S_1 - (\lambda_{E_2} - \lambda_{E_1})m_2^1 E_1 - (\lambda_{R_2} - \lambda_{R_1})m_3^1 R_1 - \\
& (\lambda_{E_2} - \lambda_{S_2})\beta_2(E_2 + I_2)S_2 - (\lambda_{I_2} - \lambda_{E_2})k_2 E_2 - (\lambda_{R_2} - \lambda_{I_2})(\sigma_2 + u_2)I_2 - \\
& (\lambda_{S_3} - \lambda_{S_2})m_1^2 S_2 - (\lambda_{E_3} - \lambda_{E_2})m_2^2 E_2 + (\lambda_{R_3} - \lambda_{R_2})m_3^2 R_2 - \\
& (\lambda_{E_3} - \lambda_{S_3})\beta_3(E_3 + I_3)S_3 - (\lambda_{I_3} - \lambda_{E_3})k_3 E_3 - (\lambda_{R_3} - \lambda_{I_3})(\sigma_3 + u_3)I_3].
\end{aligned} \tag{A5}$$

The system of ODE is autonomous, i.e., Hamiltonian, and the constraints on the control do not have an explicit dependency on the independent variable $t$

$$H(\lambda, S_1(t), Q_1(t), \dots, R_3(t), u_1(t), u_2(t), u_3(t)) = \text{const.} \tag{A6}$$

From (8)–(13), we obtain

$$\begin{aligned}
H = H(T) = {} & -(f_1(I_1(T)) + f_2(I_2(T)) + f_3(I_3(T)) + \\
& h_1(u_1(T)) + h_2(u_2(T)) + h_3(u_3(T))) \le 0.
\end{aligned} \tag{A7}$$

Since $(\lambda_{R_1}(t) - \lambda_{I_1}(t))$ is a decreasing function on the interval $[0; T]$; then, according to Property A1, we consider a time moment $t^{*+}$ such as:

$$\dot{\lambda}_{R_1}(t^{*+}) - \dot{\lambda}_{I_1}(t^{*+}) = (\lambda_{R_1} - \lambda_{R_2})m_3^1 - f_1'(I_1) - (\lambda_{S_1} - \lambda_{E_1})\beta_1 S_1 - (\lambda_{I_1} - \lambda_{R_1})(\sigma_1 + u_1). \tag{A8}$$

Substituting the function expressed from the Hamiltonian into this derivative, we get:

$$\dot{\lambda}_{R_1}(t^{*+}) - \dot{\lambda}_{I_1}(t^{*+}) = (\lambda_{R_1} - \lambda_{R_2})m_3^1 - f_1'(I_1) - (\lambda_{S_1} - \lambda_{E_1})\beta_1 S_1 + \frac{1}{I_1}[H + f_1(I_1) +$$

$$f_2(I_2) + f_3(I_3) + h_1(u_1) + h_2(u_2) + h_3(u_3) - (\lambda_{E_1} - \lambda_{S_1})\beta_1(E_1 + I_1)S_1 -$$
$$(\lambda_{Q_1} - \lambda_{S_1})\delta_1 S_1 - (\lambda_{E_1} - \lambda_{Q_1})p_1 Q_1 - (\lambda_{I_1} - \lambda_{E_1})k_1 E_1 - (\lambda_{S_2} - \lambda_{S_1})m_1^1 S_1 -$$
$$(\lambda_{E_2} - \lambda_{E_1})m_2^1 E_1 - (\lambda_{R_2} - \lambda_{R_1})m_3^1 R_1 - (\lambda_{E_2} - \lambda_{S_2})\beta_2(E_2 + I_2)S_2 -$$
$$(\lambda_{I_2} - \lambda_{E_2})k_2 E_2 - (\lambda_{R_2} - \lambda_{I_2})(\sigma_2 + u_2)I_2 - (\lambda_{S_3} - \lambda_{S_2})m_1^2 S_2 -$$
$$(\lambda_{E_3} - \lambda_{E_2})m_2^2 E_2 + (\lambda_{R_3} - \lambda_{R_2})m_3^2 R_2 - (\lambda_{E_3} - \lambda_{S_3})\beta_3(E_3 + I_3)S_3 -$$
$$(\lambda_{I_3} - \lambda_{E_3})k_3 E_3 - (\lambda_{R_3} - \lambda_{I_3})(\sigma_3 + u_3)I_3] =$$

$$\frac{1}{I_1}(-f_1'(I_1)I_1 + f_1) - (\lambda_{R_2} - \lambda_{R_1})(1 + \frac{R_1}{I_1}) + \frac{1}{I_1}[H + f_2(I_2) + f_3(I_3) +$$
$$h_1(u_1) + h_2(u_2) + h_3(u_3) - (\lambda_{E_1} - \lambda_{S_1})\beta_1 E_1 S_1 -$$
$$(\lambda_{Q_1} - \lambda_{S_1})\delta_1 S_1 - (\lambda_{E_1} - \lambda_{Q_1})p_1 Q_1 - (\lambda_{I_1} - \lambda_{E_1})k_1 E_1 - (\lambda_{S_2} - \lambda_{S_1})m_1^1 S_1 -$$
$$(\lambda_{E_2} - \lambda_{E_1})m_2^1 E_1 - (\lambda_{E_2} - \lambda_{S_2})\beta_2(E_2 + I_2)S_2 -$$
$$(\lambda_{I_2} - \lambda_{E_2})k_2 E_2 - (\lambda_{R_2} - \lambda_{I_2})(\sigma_2 + u_2)I_2 - (\lambda_{S_3} - \lambda_{S_2})m_1^2 S_2 -$$
$$(\lambda_{E_3} - \lambda_{E_2})m_2^2 E_2 + (\lambda_{R_3} - \lambda_{R_2})m_3^2 R_2 - (\lambda_{E_3} - \lambda_{S_3})\beta_3(E_3 + I_3)S_3 -$$
$$(\lambda_{I_3} - \lambda_{E_3})k_3 E_3 - (\lambda_{R_3} - \lambda_{I_3})(\sigma_3 + u_3)I_3] \tag{A9}$$

According to assumption (A3), Property A2, (21), and (A7), the difference $(\dot{\lambda}_{R_1}(t) - \dot{\lambda}_{I_1}(t))$ is negative at the time moment $t^{*+}$, then function $(\lambda_{R_1}(t) - \lambda_{I_1}(t))$ decreases. This contradicts Property A1 and proves that our function is not increasing on the interval $t \in [0; T]$. We find that $(\lambda_{R_1}(t) - \lambda_{I_1}(t)) \geq 0$ for $t \in [0; T]$. The proof for all other functions from (A3) is similar to this.

The proof of Lemma 1 is completed. $\square$

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
