# Peer review of "Hierarchical Epidemic Model on Structured Population: Diffusion Patterns and Control Policies"

_computation, doi:10.3390/computation10020031_

Round 1

Reviewer 1 Report

See attached pdf

Author Response

We would like to thank you for providing the opportunity of improving our manuscript. We also appreciate the patience, useful remarks and thoughtful comments from the reviewers.

  • We have revised and edited the text of manuscript and fixed errors, typos and the technical remarks from reviews;
  • We have reorganized the introduction and conclusion sections enhancing discussion on the possible solution methods, and added citation of the related works;
  • The structure of the paper has also been reorganized adding subsection about basic reproduction number.

“…connection to actual systems is not explored, e.g., what types of control strategies would have concave up or down cost functions and why? An explanation of parameter selection for the simulations is not given.”

Thank you very much for the comments. We supplement the formulation of the model by the explanation of the control strategies and necessary citations.

We very much appreciate your comments, however, we would like to note, that the main purpose of our work is to investigate the effect of quarantine at the first level of the hierarchical system. As part of our work, we have analyzed whether the use of quarantine and special medical measures is sufficient to prevent the development of an epidemic. We also considered different types of residents, such as responsible people who decided to stay in quarantine themselves and not to leave it, and irresponsible people who did not stay at home and did not take protective measures. As a result of the experiment, we found that the behavior of the agents of the population significantly affects the evolution of the system, which is reflected in experiments 3 and 4. We understand the necessity of the numerical simulation on real-life data. Now we are working on simulations for statistical datasets from different countries. We hope to incorporate these results into our next paper.

“There are details of the model that are not clear at the outset, but become clear as the results are developed. For instance, there is some confusion with the quarantined class, does it return to the susceptible or pass to the infected class?”

Thank you for the constructive comment. We have corrected the inaccuracy in the experiments by writing that an individual returns from Quarantine (Q) to a Susceptible state (S). This assumption was made in one of the working models. In the current case, we assume that an individual who chose not to comply with the safety requirements eventually becomes infected, moves to the Exposed state (E), and becomes a carrier of the disease.

“A more general concern of this reviewer is that what the different “levels” actually are is vague. Are there differences in the dynamics at each level? The population at each level is not conserved, only the population of the whole. Infected and susceptible move to next level at some fixed rate. In the end the bulk of the population is counted in the highest level. The structure of the model is like a bucket cascade, with the eventual mixing of all levels of the population. This is difficult to interpret in reality, it doesn’t seem reasonable that a large percentage of the population of a village, say, should eventually be in contact with the population of the entire country. This should be explained more completely in the model development.”

Thank you for your remark. We have added our motivation into the text.

This approach has been motivated by the initial stage of the COVID-19 propagation in China and Italy. The epidemic growth started in a small local village from single cases, then it was propagated to the nearest regions and the whole country. Following this idea, in the considered model, in addition to the standard formulation of an epidemic model, the whole population is split into three clusters (levels). Additionally, it is also assumed that migration between the levels is permitted. The presence of migration leads to the epidemic that started in cluster one, taking over all others. Depending on the migration and infection rates, specific restrictions can be applied to protect the population from epidemics. In analyzing different scenarios of protective measures, the control model is formulated subject to several types of the control strategies such as quarantine in the initial cluster and treatment for all clusters.

Reviewer 2 Report

This study addresses disease control from the perspective of control theory.

The spreading of disease is modeled as a three-layer structure with SEIR. The disease control is formulated as an optimal control problem. Control theory is then applied to decide the best quarantine parameters for given model parameters. Numerical results present the outcomes of different parameter settings.

This work is mathematically sound and complete.

However, there are two critical issues.

  1. Lacking a comparative study renders this work less persuasive.
  2. The feasibility of the model and the effectiveness of quarantine policy are not validated with real-world data. Without proper validation, the entire work becomes meaningless.

Author Response

We would like to thank you for providing the opportunity of improving our manuscript. We also appreciate the patience, useful remarks and thoughtful comments from the reviewers.

  • We have revised and edited the text of manuscript and fixed errors, typos and the technical remarks from reviews;
  • We have reorganized the introduction and conclusion sections enhancing discussion on the possible solution methods, and added citation of the related works;
  • The structure of the paper has also been reorganized adding subsection about basic reproduction number.

 “The feasibility of the model and the effectiveness of quarantine policy are not validated with real-world data.”

Thank you very much for the comments. We supplement the formulation of the model by the explanation of the control strategies and necessary citations.

We very much appreciate your comments, however, we would like to note, that the main purpose of our work is to investigate the effect of quarantine at the first level of the hierarchical system. As part of our work, we have analyzed whether the use of quarantine and special medical measures is sufficient to prevent the development of an epidemic. We also considered different types of residents, such as responsible people who decided to stay in quarantine themselves and not to leave it, and irresponsible people who did not stay at home and did not take protective measures. As a result of the experiment, we found that the behavior of the agents of the population significantly affects the evolution of the system, which is reflected in experiments 3 and 4. We understand the necessity of the numerical simulation on real-life data. Now we are working on simulations for statistical datasets from different countries. We hope to incorporate these results into our next paper.

Lacking a comparative study renders this work less persuasive”.

 We have added more articles to our references and comparative analysis to our experiments.

Reviewer 3 Report

Let we assume that the model is fine and interesting however, there is lacking of mathematical and numerical results. I can suggest major revision,

  1. Provide background mathematical results for the main model.
  2. What are the equilibrium points of the model?
  3. Where is the basic reproduction number for each case of the model (submodel)?
  4. How can directly the model for the control is formulated without discussing the sensitivity analysis.
  5. I cannot see any supported data for the numerical results.
  6. Is there any data that can support the numerical findings given in Figures?
  7. i do not agree with the optimal control results, it should be revised with care, by providing the parameters values as well as the model variables with initial conditions.
  8. The finding of the optimal controls graphs are not sufficient for this study.
  9. I can suggest the related reference that can be useful for this study and introduction part as well.
  10. A dynamical model of asymptomatic carrier zika virus with optimal control strategies Nonlinear Analysis: Real World Applications 50, 144-170.

Author Response

We would like to thank you for providing the opportunity of improving our manuscript. We also appreciate the patience, useful remarks and thoughtful comments from the reviewers.

  • We have revised and edited the text of manuscript and fixed errors, typos and the technical remarks from reviews;
  • We have reorganized the introduction and conclusion sections enhancing discussion on the possible solution methods, and added citation of the related works;
  • The structure of the paper has also been reorganized adding subsection about basic reproduction number.

Provide background mathematical results for the main model”.

We have added more articles to our references.

“What are the equilibrium points of the model?”

In current model, the equilibrium points are in the cases where there are no infected individuals in the system and all others are in Susceptible (S) or Recovered (R) states. We decided not to include this information in the article because it is irrelevant information.

“Where is the basic reproduction number for each case of the model (submodel)?”

We have added a part with calculation of the basic reproduction number R0 in Section 2 and added the relevant articles to the list of references, i.e. Diekmann O., Heesterbeek. A. P., Roberts M.G.  The construction of next-generation matrices for compartmental epidemic models; Van den Driessche P. Reproduction numbers of infectious disease models.

“How can directly the model for the control is formulated without discussing the sensitivity analysis.”

Our main purpose of the article was to construct the controlled model with control measures as quarantine and treatment of the infected individuals and estimate the effect of the controlled parameters on the aggregated costs. However, we respectfully believe that according to a vast range of the research studies devoted to epidemic processes and optimal control, listing of the examples of optimization problems elongate the paper and can be easily found in the reference literature, i.e. Altman E., Khouzani M. and Sarkar S.  “Optimal control of epidemic evolution”. The application of the sensitivity analysis to our problem provides the results which can be the subject for further research.

“I cannot see any supported data for the numerical results. Is there any data that can support the numerical findings given in Figures?”

Thank you very much for the comments. We supplement the formulation of the model by the explanation of the control strategies and necessary citations.

We very much appreciate your comments, however, we would like to note, that the main purpose of our work is to investigate the effect of quarantine at the first level of the hierarchical system. As part of our work, we have analyzed whether the use of quarantine and special medical measures is sufficient to prevent the development of an epidemic. We also considered different types of residents, such as responsible people who decided to stay in quarantine themselves and not to leave it, and irresponsible people who did not stay at home and did not take protective measures. As a result of the experiment, we found that the behavior of the agents of the population significantly affects the evolution of the system, which is reflected in experiments 3 and 4. We understand the necessity of the numerical simulation on real-life data. Now we are working on simulations for statistical datasets from different countries. We hope to incorporate these results into our next paper.

“I do not agree with the optimal control results, it should be revised with care, by providing the parameters values as well as the model variables with initial conditions.”

The optimal control structure was determined using the Pontryagin maximum principle. A similar method was used in Altman E., Khouzani  M.,  Sarkar S.  Optimal control of epidemic evolution (2011); Taynitskiy V., Gubar E., Zhu Q. Optimal Control of Heterogeneous Mutating Viruses (2018). In the course of the work, the necessity and sufficiency of optimal control was shown. The proof of the method can be found in the Appendix.

The structure of the optimal control is determined by considering the various parameters of the system and the initial data.

“The finding of the optimal controls graphs are not sufficient for this study.”

Optimal control graphs show a treatment plan that helps reduce system costs. The graphs are constructed according to the structure of optimal control. Therefore, we consider these graphs important in our article.

 “I can suggest the related reference that can be useful for this study and introduction part as well.”

Thank you very much for the suggested article, we have studied it and added to our list of references.